# A New Insight into High-Aspect-Ratio Channel Drilling in Translucent Dielectrics with a KrF Laser for Waveguide Applications

**DOI:** 10.3390/ma15238347

**Published:** 2022-11-24

**Authors:** Igor V. Smetanin, Alexey V. Shutov, Nikolay N. Ustinovskii, Polad V. Veliev, Vladimir D. Zvorykin

**Affiliations:** P. N. Lebedev Physical Institute, Leninskii Prospekt, 53, 119991 Moscow, Russia

**Keywords:** multi-pulse drilling of dielectric materials by KrF laser, laser beam filamentation and waveguide propagation in long capillary channel

## Abstract

A new insight into capillary channel formation with a high aspect ratio in the translucent matter by nanosecond UV laser pulses is discussed based on our experiments on KrF laser multi-pulse drilling of polymethyl methacrylate and K8 silica glass. The proposed mechanism includes self-consistent laser beam filamentation along a small UV light penetration depth caused by a local refraction index increase due to material densification by both UV and ablation pressure, followed by filamentation-assisted ablation. A similar mechanism was shown to be realized in highly transparent media, i.e., KU-1 glass with a multiphoton absorption switched on instead of linear absorption. Waveguide laser beam propagation in long capillary channels was considered for direct electron acceleration by high-power laser pulses and nonlinear compression of excimer laser pulses into the picosecond range.

## 1. Introduction

An effective ablation of various polymer materials by excimer lasers was first demonstrated by Kawamura et al. [1], Srinivasan and Mayne-Banton [2], and Andrew et al. [3] shortly after the advent of commercially available excimer lasers. Excimer lasers generate high-power pulsed radiation in the UV and VUV spectral ranges with wavelengths *λ* = 157–351 nm, just where polymers show a rapid growth of absorption coefficient *α*(*λ*) [4]. These pioneering works formed the basis of a new direction in the microstructuring and micromachining of polymers and are widely used in microelectronics, microscale fluidic devices, medical applications, etc. Over the past 40 years, several reviews and books have been published in addition to original works on this important topic (e.g., see [5,6,7,8,9,10,11,12]). Most of them concern “surface ablation”, when the ablated depth is small compared with transverse structure dimensions. Moderate laser fluences are typically beyond the ablation threshold and a limited number of consequent laser pulses. In this paper, we will give a qualitative description of the physical processes underlining the interaction of UV laser radiation with polymers, which is an important starting point for the present work in terms of the “bulk ablation” in deep laser-drilled channels. The latter phenomenon will be reviewed in the rest of the introduction.

The low thresholds of the UV ablation of polymers are due to the low binding energy of polymer molecular chains compared to the energy of UV laser photons, i.e., 7.9 eV (for F_2_ laser), 6.4 eV (ArF), 5.6 eV (KrCl), 5.0 eV (KrF), 4 eV (XeCl), and 3.5 eV (XeF). The breaking of these covalent chain bonds leads to a photochemical decomposition of solid polymers with a large-number initial molecular weight into average-weight monomers and low-weight molecular fragments. These are volatile and promptly expel with a supersonic velocity from the bulk polymer volume affected by the laser light along its penetration depth *l_eff_* = 1/*α_eff_*. An effective absorption coefficient *α_eff_*(λ) might differ from a linear absorption coefficient *α*(*λ*) determined at a low light intensity, as the matter in front of the ablation front could be heated and modified by a leading part of the laser pulse or by previous pulses in the case of a rep-rate laser operation [10]. This mechanism of so-called “photochemical etching” [2,13] gives the etch depth (rate) *l* vs. laser fluence *F* dependence which follows from the Beer–Lambert law with exponential attenuation of the light in the solid matter:(1)$$l_{ab}=\frac{1}{\alpha_{eff}}ln\left(\frac{F}{F_{th}}\right)$$

Here *F_th_* is a threshold fluence value for the ablation onset. It is stated that for *F* < *F_th_*, there is no material removal at all while the absorbed laser fluence is transformed into heat and causes a sub-threshold material modification known as the “incubation” [14,15]. On the other hand, the threshold fluence *F_th_* can be expressed through the ablation enthalpy of the material *H*, which is the minimal volumetric energy density required for a solid transformation into volatile products and laser beam penetration depth by the approximate formula:(2)$$F_{th}=\eta H{\left(\alpha_{eff}\right)}^{-1},$$
where *η* ≤ 1 is a quantum yield of a direct scission of chain bonds determined by the kinetics of the photoablation process.

It means that the ablation rate and threshold fluence are inversely proportional to the absorption coefficient for various materials. By combining (1) and (2), one can see that for a given material, there is an optimal *α_eff_*(λ) corresponding to a maximal ablation depth, while in fully transparent materials with α*_eff_* → 0, the ablation depth *l_ab_* → 0. In other words, any semitransparent material in the UV–VUV range, like polymers or glasses, can be effectively treated by excimer lasers with a properly chosen wavelength. For instance, KrF laser radiation produces the maximal ablation rate of polymethyl methacrylate (PMMA), which has rather low absorption at *λ* = 248 nm compared with other polymers and threshold fluence *F_th_* = 0.1–0.3 J/cm^2^ [16,17,18]. Another way to tune absorption to optimal value is polymer doping by a small amount of appropriate substances or dyes that increase the ablation rate without changing the other polymer properties [16]. 

In contrast to surface ablation, deep reliefs at the polymer surface, as well as microholes with a very high aspect ratio of the length to diameter *A* = *L*/*D* were produced in a multi-pulse regime at increased laser fluence and irradiation times [19,20,21,22,23]. Such capillary-like channels drilled in various polymer materials by a KrF laser with an energy of up to 350 mJ and pulse duration of 25 ns are the most related to our present experiments. A uniform sample irradiation was obtained with a two-lens projection scheme with a laser fluence varied in the range *F* = 0.5–50 J/cm^2^ in focal spots of 2–100 µm, which corresponded to maximum intensities in the pulse *I* = 2.5 × 10^7^–2.5 × 10^9^ W/cm^2^. The longest capillary-like channel of about 20 mm in length and the inlet diameter of 35 μm (*A* ≈ 600) was obtained in poly(ethylene terephthalate) (PET). PET has a high absorption coefficient *α* = 16 µm^−1^ corresponding to radiation penetration depth *l* = 0.065 µm, a low ablation threshold *F_th_* = 30 mJ‧cm^−2^, and it was ablated with the rate *l_ab_* = 0.7 µm/pulse. 

For comparison, more transparent PMMA with *α* = 0.063 µm^−1^, *l* = 150 µm, *F_th_* = 250 mJ‧cm^−2^ was drilled with a fourfold higher rate *l_ab_* = 2.5 µm/pulse; the maximum channel length was about 10 mm and aspect ratio *A* ≈ 300. The dynamics of channel formation in all polymers were similar: their growth rate, the highest in the beginning, then gradually slowed down to a constant value, which did not depend on *F*. When the growth stopped, and the channel reached its final length *L_max_*, it had a nearly conical shape with an apex angle of ~ 1/*A*. The transmitted radiation energy decreased linearly with the channel length due to interaction with the wall, and an even smaller amount reached the bottom. According to formula (1), the channel growth rate should have decreased logarithmically, but it remained constant over its entire length until the ablation stopped. This is the first issue to be clarified in the present work. The theoretical model [21,22,23] considered the channel shape formed self-consistently due to the transport of laser radiation diffusely scattered on the rough wall. As observed in similar experiments (see, e.g., [24]), the possibility of a waveguide propagation regime was unreasonably ignored. As the role of specular reflection does increase at grazing incidence angles of radiation onto the wall inside a long channel, this assumption should also be clarified. An interplay explained the constant ablation rate of the growing channel between radiation absorption in the wall and the ablation plasma adjacent to the channel bottom. 

In our opinion, although the mechanisms of formation of deep channels in semitransparent dielectrics—particularly in polymeric materials—under the action of UV radiation from excimer lasers have been widely studied for 40 years, they are still not clear. Therefore, it was a goal of the present experiments to locate the true mechanisms of high-aspect-ratio capillary channel formation in typical materials, such as PMMA and translucent fused silica glass K8 (which is an analog of Schott Glass BK7 glass) for a KrF laser in comparison with the drilling of highly transparent KU-1 glass (analog of Corning 7980 glass). Earlier, we observed extended channels with a length of up to 1 mm and a diameter of ~30 μm under entirely different conditions, namely in the interaction of a single high-power KrF laser pulse of 100 J energy and 100 ns duration with PMMA targets [25,26,27,28]. At a much higher laser intensity, *I* ~ 5 × 10^12^ W/cm^2^, a channel was produced in the PMMA due to pre-focusing of radiation that passed through a hot (*T* ~ 50 eV) dense (*ρ* ~ 1 g/cm^3^) ablation plasma and was reflected at a grazing angle by the wall of a deep conical crater with an inlet diameter *D* ~ 0.5 mm and depth *L* ~ 1 mm. 

A conical shock wave was also seen in microscope images obtained in polarized light, which evidenced an increased refraction index in the target material densified by the ablation pressure. A narrow channel extended from the crater cone apex inside the target was assumed to be formed by a self-focusing laser radiation in the target material. It was proved by our experiments on multi-pulse ablation of semitransparent materials, in particular, PMMA and K8 glass. A multi-track ablation pattern in these materials, especially glass, looked like it was caused by multiple filaments of the laser beam (see, e.g., [29,30]). Indeed, for fused silica glass at KrF laser wavelength *λ* = 248 nm linear refraction index is *n*_0_ = 1.5 while the nonlinear one due to the Kerr effect is *n*_2_ = (3.4–5.6) × 10^−16^ cm^2^/W [31,32]. Those give the critical power for radiation self-focusing $P_{cr}=3.77\lambda^{2}/8\pi n_{0}n_{2}$ = (1.1–1.5) × 10^5^ W, which is much lower than the peak power *P* ~ 10^9^ W in high-energy single-shot experiments [25,26,27,28]. Moreover, it was comparable with *P* ≥ 10^5^ W in small-energy multi-pulse experiments. The question remains whether self-focusing can be developed along a limited penetration depth of KrF laser radiation ahead of the ablation front in a condensed matter. In K8 glass, it is *l ~* 30 μm; while experimental data for PMMA are rather contradictory, from a few to the hundreds of microns, probably depending on the sample preparation technique [4,16,18,23].

In this paper, we first provide additional arguments in favor of the decisive role of self-focusing in the multi-pulse moderate-power ablation of translucent materials. Secondly, we demonstrate the waveguide nature of laser radiation transport within long channels. Among other things, the latter can be responsible for a direct acceleration of plasma electrons in the channel by the electric field of laser light [33]. Finally, the possibility of the effective shortening of UV laser pulses in the channel over a large nonlinear interaction length of laser radiation with SBS and SRS active media will be addressed.

## 2. Experimental Setup

For drilling capillary-like channels in translucent dielectric materials, a commercial discharge-pumped KrF laser (EMG 150 TMSC model, Lambda Physik) was used, which operated with maximum energy of *E_Lmax_ ≈* 200 mJ in 20-ns pulses (at FWHM) with 10 Hz rep-rate and beam divergence ~0.2 mrad. The layout of the sample-drilling experiments is shown in Figure 1a. Laser radiation was attenuated to the dynamic range *E_L_* = (0.01 ÷ 0.94)*E_Lmax_* by a stepwise diffraction attenuator DVA-22-250 (Institute of Automation and Electrometry, Siberian Branch of RAS) and focused on the sample surface with a lens with a focal length *f* = 100 cm. A Canon 80 D camera registered the channel drilling process with a 100 mm f/2.8 lens with a frame frequency of 24 per second through a transparent side wall of the sample. The rep-rate drilling process was finished when the channel stopped growing or came out through the whole sample thickness. We further investigated the waveguide properties of the drilled channels. Spatial distributions in the near field behind the channel (Figure 1b) and in the far field immediately at the channel outlet (Figure 1c) were measured with a Spiricon SP-620U beam profiler (Ophir Photonics); the calorimeter PESO-SH-V2 measured the transmitted energy through the channel with NOVA II display (Ophir Photonics). 

To measure the UV light distribution, the required laser radiation was imaged onto the K8 glass plate that converted UV into green fluorescence, which was collected by the L3 lens onto the beam profiler CCD matrix (Figure 1b,c). Then the distribution of laser fluence *F* was found from the fluorescence fluence *F_fl_* using the calibration dependence $F\propto F_{fl}^{2.5}$ [28].

The distribution of incident radiation in the focal spot being measured with the same method with the K8 glass converter had a Gaussian-like symmetric central part of 60-μm diameter at the FWHM, and broad vertically elongated low-intensity wings originated from a temporal evolution of the laser light in the unstable resonator cavity (Figure 2). For the limited pumping time, only a few round trips of laser light in the resonator had time to occur, each with successively decreased output beam divergence. As a result, the rectangular laser aperture was reproduced in the far-field zone during the first-round trip [28]. 

For a given incident’s laser energy *E_L_* and measured fluence distribution, the maximum fluence in the center of the focal spot *F*_0_ could be expressed as *F*_0_ [J/cm^2^] = 5 × 10^3^
*E_L_* [J]. For the pulse width *τ_L_* = 20 ns corresponding peak intensity *I*_0_ was *I*_0_ [W/cm^2^] = 2.5 × 10^11^
*E_L_* [J]. For the maximum energy *E_Lmax_* = 200 mJ these give *F*_0_ = 1 kJ/cm^2^ and *I*_0_ = 50 GW/cm^2^. Note that the present Gaussian-like distribution of incident radiation strongly differs from the uniform one in experiments [20,21,22,23].

## 3. Experimental Results

### 3.1. PMMA Absorption in UV Spectral Range

PMMA is chosen as a basic material and has been intensively studied. Nevertheless, as mentioned above, there is a vast spectrum of absorption coefficient values for PMMA in the UV spectral range in the literature. For this reason, we measured with spectrophotometer dependences α(λ) for colorless and dyed PMMA samples used in drilling experiments. They are shown in Figure 3. As transmittances of 1 mm thick samples rapidly fall to zero for λ < 300 nm (see Figure 3a), thinner PMMA samples of ~100 μm thickness were prepared for precise measurements. A sandwich-like structure containing a PMMA layer was produced by heating a small PMMA piece to plasticity and squeezing it between two flat UV-visible transparent fused-quartz plates. The obtained sandwich was measured at the spectrophotometer Specord M40 and compared with the transmittance of the quartz plates alone. The Fresnel reflection at the interfaces of PMMA, air gaps and quartz was considered. It is seen in Figure 3b that at the KrF laser wavelength λ = 248 nm, the absorption coefficient in the colorless PMMA α =120 cm^−1^ (penetration depth l = 80 μm) is less than half that of the green one. 

### 3.2. Drilling of Translucent and Transparent Materials in a Multi-Pulse Regime

In this section, the results of drilling some semi-transparent and transparent materials with different penetration depths are presented, namely colored and dyed PMMA, as well as K8 and KU-1 glass, by multiple pulses of KrF laser. PMMA was chosen as the basic material, as it had been intensively studied. K8 glass with a large ablation threshold and a penetration depth *l* ≈ 30 μm was compared with pure fused silica glass KU-1, which has a minimal linear UV absorption coefficient *α* (*λ* = 248 nm) ≤ 0.002 cm^−1^ (penetration depth *l* ≈ 5 m). 

#### 3.2.1. Drilling of PMMA

Selected frames from the video camera records of the PMMA drilling shown in Figure 4 demonstrate the main steps of channel formation (see also Appendix A). Laser radiation falls onto a sample from the right. The left-hand-side column corresponds to a 10 Hz drilling run with relatively low pulse energy *E_L_* ≈ 2.8 mJ (*F*_0_ = 5.6 J/cm^2^). In this case, a linear growth of the channel length *z* with time was observed from the beginning up to its approach to the final length *z_max_* ≡ *L_max_* ≈ 6 mm, at which point the ablation rate slowed to zero. The inlet channel diameter *D_in_* was about a few hundred microns, as determined by the focal spot’s energy distribution. Only a high-intensity central part of incident radiation energy *ζE_L_* participated in the channel drilling, while low-intensity wings produced a surface sample ablation. In the second drilling run of the same hole when the pulse energy was increased by 2.5 times (see a right-hand column in Figure 4), the channel continued to grow with the same velocity up to *L_max_* ≈ 20 mm, and was somewhat thicker (at the same depth *z*) than in the first run. When *z → L_max_*, the channel end had a diameter *D_fin_* of a few tens of microns (Figure 5). The brightest luminosity of plasma in the channel was observed in a narrow region Δ*z* ~ *D*(*z*) adjoined to the moving ablation front. As a rule, towards the end, the single channel suddenly split into many, then only a few of them survived and continued to grow to the final length *L_max_* while the rest stopped.

In more detail, some of the channel features in PMMA can be seen in microscopic images of the fully completed channels (Figure 5). At a low laser pulse energy *E_L_* =0.3 mJ, i.e., below the threshold of linear channel growth, the ablation crater has a cone-like form ended by a short channel of ~ 100 μm length (a). The channel emerges from the cone tip with a diameter of ~ 10 μm at the end, which is by an order of magnitude less than the focal spot diameter. It was supposedly formed by pre-focusing the incident radiation in the course of its specular reflection at grazing angles by the crater wall that were followed by nonlinear self-focusing along the penetration depth in PMMA. The crater and channel structure in the present multi-pulse low-energy drilling experiments are very similar to those obtained in a high-energy single-pulse interaction with the PMMA [25,26,27,28].

At a high energy of *E_L_* =22.3 mJ, when a long channel has been completely formed, a well-defined splitting of the channel into multiple filament-like tracks of ~ 10 μm in diameter was observed in the final stage that deviated from the beam axis (b). This evidently indicates the instability of the ablation front just before the channel growth stopped.

Figure 6 shows a set of dependencies of channel length *z* vs. drilling time *t* for different laser pulse energies at a 10 Hz rep rate. Each channel was drilled separately, starting from the sample surface. It is seen that for *E_L_* ≥ 2.8 mJ, all dependencies *z*(*t*) had the same linear growth rate before the onset of saturation that appeared at the channel length designated as *L_lin_*, which is slightly shorter than the maximal length *L_max_*. 

As seen in Figure 7, *L_lin_* logarithmically depends on the pulse energy *E_L_*. This might be explained by an exponential decrease in laser energy along the channel length
(3)$$E\left(z\right)=\zeta E_{L}exp\left(-\beta z\right)$$
where *ζE_L_* is a part of the incident pulse energy captured into the channel and *β* is the attenuation coefficient along the channel length. From the present experiment, it can be estimated that *β* ≈ 0.085 mm^−1^, assuming that *ζ* is the same for different incident energies, while *L_lin_* is associated with some minimal transferred energy required for the ablation.

Figure 8 shows *z*(*t*) dependencies obtained for the colorless and dyed PMMA samples in the linear ablation range before reaching saturation. It is seen that they fully coincided for all PMMA samples, despite a significant difference in absorption coefficients and ablation rates that might be expected from formula (1). From a slope of the linear *z*(*t*) dependence in Figure 6 and Figure 8, the ablation rate of *l_ab_* =3.5–4 μm/pulse was found for the colorless and dyed PMMA samples. Let us emphasize that in contrast to a surface material ablation (see the Introduction), the channel growth rate was constant during the drilling time, independent of the pulse energy and penetration depths of laser radiation into the PMMA samples. 

#### 3.2.2. Drilling of K8 Glass

The instability of the ablation front, which was manifested in the PMMA by a fork-like channel end, appeared in the K8 glass in the form of multiple filaments. They are well-defined at the near-threshold pulse energy *E_L_* ≈ 2–4 mJ (*F*_0_ = 10–20 J/cm^2^) in Figure 9a. As can be seen, only one filament (designated by 1) survived for time. It was relatively straight along the length *L_max_* ≈ 3 mm and had a diameter *D* ~ 10 μm corresponding to an aspect ratio *A ≈* 300. A cylindrical shell (2) can also be seen surrounding the channel. It had a diameter of ~ 100 μm and apparently originated from the glass densification by the ablation pressure. Generally, material compressed by ablation pressure has an increased refraction index. Such a coaxial structure of a through-hole surrounded by a zone with increased refractive index, both of which are in the volume of unmodified material, possess the property of a waveguide structure when transmitting laser light [34,35] Meanwhile, a bundle of initial filaments (3) formed nearby the surface and stopped growing.

A single, thick channel was drilled when the pulse energy was increased up to ~ 10 mJ (Figure 9b). Since the drilling stopped long before the channel reached its final length, the channel form was not very sharp, and a thin microstructure was discernible at its end. The channel asymmetry was likely caused by a slight inclination of the sample surface to the incident radiation. The drilling rate was found to be *l_ab_* ≈ 0.35–0.45 μm/pulse, i.e., about tenfold less than in PMMA.

#### 3.2.3. Drilling of KU-1 Glass

In contrast to the translucent materials (PMMA and K8 glass), in highly transparent samples of high-purity fused silica glass KU-1 with absorption coefficient *α* ≤ 0.002 cm^−1^, at low pulse energy *E_L_* ≤ 1 mJ, there was no damage on the surface nor in the bulk. At *E_L_* ≈ 2 mJ, several luminous foci arose simultaneously in the bulk material along the laser beam (see Figure 10 and Appendix A). The beaded structure, which was tens millimeters long, appeared just after the first few laser pulses with a minimal change during the next series of pulses. It was obviously associated with an optical breakdown in the bulk material due to the self-focusing of radiation. Discrete foci, however, did not form a continuous channel. However, a continuous channel similar to that in the K8 glass grew inward from the irradiated surface (Figure 11a) with the drilling rate *l_ab_* ≈ 0.03 μm/pulse. For higher pulse energies *E_L_* ≈ 5 and 20 mJ, the optical breakdown gradually became faint and disappeared; the higher the energy, the thinner the channel became. The channel grew with approximately the same rate *l_ab_* ≈ 0.06 μm/pulse, which is less than in the translucent K8 glass by an order of magnitude. 

### 3.3. Laser Beam Transportation through Drilled Channels

#### 3.3.1. Channel Transmittance

For channels drilled in samples of various thicknesses, energy transmittance was directly measured by calorimeters, as shown in Figure 1. These measurements were carried out for PMMA with a thickness of 1 to 23 mm, immediately after the completion of the drilling process at incident energy *E_L_* ≈ 10 mJ when the channel went to the back of the sample. The transmittance of the channels decreased with the length, as shown in Figure 12. Assuming that for a fraction *ζ* = 0.4 of incident energy *E_L_* captured into the channel, one can observe that the measured dependence is well-approximated by an exponential decrease of the transferred laser radiation in the channel with an attenuation coefficient *β* = 0.085 mm^−1^ in agreement with Section 3.2.1.

#### 3.3.2. Spatial Distribution of Laser Radiation in the Channel

The spatial distribution of the laser radiation after the passage of channels drilled in the PMMA with different thicknesses was measured at approximately the same laser pulse energy of 15 mJ using the UV-to-green converter, which was located at 100 mm from the channel outlet, which allowed for the formation of the mode structure in radiation passed through the channel (Figure 13) to be observed. 

In a special case when there was no sample (a), the relatively uniform distribution was observed in the shape of a rectangle corresponding to a zone near the laser beam diverging after the focus. The radiation was transmitted through short-length channels and had a similar distribution in the thin samples with a thickness of ~ 1 mm. As the sample thickness (i.e., the channel length) increased, “hot spots” appeared in the rectangular distribution, from which several spatial modes were then formed (b, c). The random arrangement of modes, varying from pulse to pulse, was apparently due to the instability of the focal spot position relative to the channel inlet caused by a slight shaking of the optical scheme mirrors and optical perturbations of the laser gain medium during the discharge pumping. The most stable mode was formed in the channel with the longest length of 23 mm, which was only a single channel (d). Its divergence was much less than the geometric divergence of the laser beam, given by the focusing lens (a).

The distribution of output laser radiation immediately behind the long channel was measured by imaging its outlet onto the UV-to-green converter, as shown in Figure 1. Although an order of magnitude increased the laser pulse energy in this experiment up to *E_L_* ≈ 110 mJ, no change in the single-mode distribution of the output radiation was observed (Figure 14), which indicates a negligible influence of the plasma formed in the channel on its waveguide properties.

## 4. Discussion: Self-Focusing and Waveguide-Like Transportation of UV Laser Radiation Govern Efficient Drilling of Capillary Channels in Translucent Dielectric Materials

Experimental data reported in Section 3.2 undoubtedly evidence that radiation self-focusing underlies the formation of capillary-like channels with a high-aspect ratio in various materials. Below, we explain our vision as to how this mechanism works with some peculiarities in both semitransparent and transparent laser light materials. 

The self-focusing of high-power laser radiation in the transparent matter was discovered about 60 years ago and was explained as a nonlinear dependence of the material refraction index on laser intensity *n*(*I*) [29,30]. It might be caused by the Kerr effect, which is the polarization of atoms and molecules in a strong laser field or other reasons for a matter response, which slightly increases *n* compared with its undisturbed (linear) value *n*_0_. It is commonly assumed that when the phase shift of the wave is due to a difference $\Delta n=n-n_{0}$ acquired along the propagation length *L_sf_* reaches 2π, a significant distortion of spatially coherent wavefront occurs. Self-focusing appears for laser beams with perfect Gaussian or irregular intensity distributions.

An appropriate criterion generally called *B*-integral:(4)$$B=\frac{2\pi }{\lambda }\int_{0}^{L_{sf}}\Delta ndl$$
should be *B* ~ 1 for the self-focusing onset. The peak intensity in the irradiation spot for incident laser energy *E_L_* = 2.8 mJ, which corresponds to the beginning of a steady-rate ablation front propagation in PMMA independent of laser energy (see Section 3.2.1), peak intensity in the irradiation spot is *I*_0_ = 7 × 10^8^ W‧cm^−2^. For *n*_2_ ≈ 4.5 × 10^−16^ cm^2^/W, like that in fused silica glass, the Kerr self-focusing mechanism gives Δ*n* = *n*_2_*I*_0_ ≈ 3 × 10^−7^. In PMMA, even if one assumed no absorption, the self-focusing length would be *L_sf_* ~ 10 cm, i.e., much longer than the penetration depth *l* ~ 100 μm of laser radiation.

Therefore, for self-focusing to be accomplished, other than the Kerr mechanism (or in addition), the refraction index variation should be looked for, providing a significantly higher Δ*n* response. A variety of UV-induced processes has been identified in PMMA [36]. Depending on the absorbed irradiation dose, they accumulate by forming a refractive index profile around the absorption zone. For example, a side chain cleavage from the main polymer chain at lower UV irradiation produces mechanical densification of the PMMA due to Van der Waals forces with a subsequent increase in the refractive index. On the contrary, the scission of the main polymer chain at higher irradiation produces a partial defragmentation of the polymer structure, decreasing the refraction index. At even higher irradiation, a total PMMA decomposition into low molecular weight fragments, namely ablation, decreases the refraction index. A complicated interplay of these processes and material densification from the ablation pressure forms a longitudinal refraction index profile *n*(*z′*) ahead of the ablation front, which varies with the absorbed radiation dose. In a multi-pulse irradiation *n*(*z′*), the distribution nonmonotonically varies from the maximal value *n_max_* ≈ 1.498 close to the ablation front (*z′* → 0) to its initial value *n*_0_ =1.4915 at *z′* > *l*. Assuming a mean refraction index difference <Δ*n*> ~ 5 × 10^−3^ over the UV penetration depth *l* ~ 100 μm in PMMA, one obtains for the characteristic self-focusing a length of *L_sf_* ~ 8 μm, which is quite enough for the laser beam filamentation to occur in PMMA. As the refraction index is proportional to material density, another reason for its growth is material compression by the ablation pressure, which works even without the photochemical transformation of PMMA by UV light. We observed the strong refraction of polarized probe light on the density gradients produced by a shock wave (which allowed us to visualize it) in high-energy experiments on the KrF laser interaction with PMMA [26,27,28], as mentioned in the Introduction.

In addition, UV-induced densification is a well-known effect in fused silica. It is caused by color center generation in the bulk material similar to other types of ionizing radiation, e.g., fast electrons, gamma rays and neutrons [37,38,39]. Together with the material densification by the ablation pressure, it produced a cylindrical dress around the filament in K8 glass (see Figure 9a), while the increased refraction index ahead of the ablation front was responsible for the UV light self-focusing along the penetration depth *l* ~ 30 μm. 

Therefore, despite different underlying processes, we believe that the UV-induced densification of translucent dielectrics is the general mechanism for a self-consistent laser beam filamentation observed in these experiments with nanosecond UV laser pulses. It is the filaments that are responsible for the ablation of these materials. In highly transparent matter such as KU-1 glass, the nonlinear processes mainly contribute to radiation absorption [40], followed by laser beam filamentation via material densification. On the other hand, Kerr self-focusing produces a prolonged multi-bead optical breakdown structure competing with continuous channel drilling. It should be noted that the high-intensity femtosecond laser pulses focused on a transparent sample perfectly drill the material due to efficient multiphoton absorption combined with radiation self-focusing [41,42,43], which is in the same manner as nanosecond pulses the translucent ones in present experiments.

The waveguide-like radial refraction index distribution around a capillary-like long channel is required for the low-loss transportation of laser radiation to the ablation front. It is formed in the translucent dielectrics due to the material densification caused by a shock wave produced by the ablation pressure, as well as by UV radiation partially penetrating through the capillary wall and locally increasing the refraction index [44]. 

Radiation self-focusing in translucent materials breaks a laser beam into a multitude of self-organized filaments carrying approximately equal power *P_f_* ~ *P_cr_* [29,30]. Therefore, a bundle of filaments, each delivering fluence *F_f_* ≈ *P_cr_*/(*πd_f_*^2^/4) where *d_f_* is the filament diameter, are drilled in parallel into the given material with the ablation rate independent of the incident laser pulse energy or power (in terms of self-focusing). The number of filaments *N_f_* (*z*) ≈ *P*(*z*)*/P_cr_* decreases gradually with radiation attenuation along the channel until the pulse power becomes equal to the critical one *P*(*z* = *L_lin_*) ≈ *P_cr_*. This implies the end of the linear channel growth. For *z > L_lin_*, the filamentation is turned off as *P*(*z*) *< P_cr_*, and without the self-focusing average radiation fluence across the irradiation area, becomes much lower than it was in the filaments, resulting in a slowdown of the ablation rate. Finally, the transmitted power and corresponding fluence fall below the ablation threshold *F*(*z* = *L_max_*) ≈ *F_th_*; thus, the channel attains its maximal length *z = L_max_*.

The experimental dependence *L_lin_* (*E_L_*) in Figure 7, together with channel transmittance measurements (Figure 12), gives the values of *ζ* = 0.4 and *β* = 0.085 mm^−1^ in the approximation formula (3), which for the evaluation of *P_cr_* in PMMA. For any incident *E_L_*, along the linear channel length *L_lin_*, the pulse energy is attenuated down to *E* (*L_lin_*) ≈ 0.5 mJ and thus corresponds to *P_cr_ ≈ P*(*z* = *L_lin_*) = *E* (*L_lin_*)/*τ_L_* = 2.5 × 10^4^ W. The peak fluence and intensity in filaments with the diameter *d_f_* = 10–20 μm is estimated to be *F_f_* = 150–600 J/cm^2^ and *I_f_* = (0.75−3.0) × ‧10^10^ W/cm^2^, which are comparable with the initial values on a sample surface at the maximal pulse energy 200 mJ. As *F_f_* is much higher than *F_th_* in PMMA and any translucent material (see Introduction) filament-assisted drilling of high-aspect-ratio channels proceeds with a relatively high ablation rate. 

## 5. Application of Capillary Channels for Nonlinear Compression of Excimer Laser Pulses and Temporal Profiling

There are several opportunities for the application of UV KrF lasers requiring a specific pulse form. For example, the inertially confined fusion with thermonuclear fuel ignition by a strong shock wave (SI ICF) demands a laser pulse of 10–20 ns length and a very sharp rise of power by about two orders of magnitude during the last 200 ps [45]. The production of long, weakly ionized plasma channels in atmospheric air by UV laser intended for the triggering and guiding of high-voltage (lightning) discharges [46], or directed transfer of microwaves along plasma waveguides [47], are the most effective using rather long high-energy pulses of hundred nanoseconds length superimposed with high-power subnanosecond pulses. These pulse forms can be obtained by combining short (*τ_sh_* ≤ *τ_c_*) and long (*τ_long_* ≥ *τ_c_*) pulses when they are simultaneously amplified in KrF gain medium with the gain recovery time *τ_c_* ≈ 2 ns [48].

Our previous experience with GARPUN, a hybrid Ti:Sapphire/KrF laser facility [49], revealed the benefits of such combined amplification and the drawbacks when operating with a rather high peak power ~1 TW in subpicocecond pulses [50]. For these goals, a more promising method seems to include the nonlinear shortening of typical 20 ns pulses of a discharge-pumped KrF laser by Stimulated Brillouin scattering (SBS) or Stimulated Raman scattering (SRS). Backward SBS and SRS schemes allow for effective pulse reflection and shortening to be obtained with sufficiently high energy conversion [51,52,53,54].

Based on the recent results of Yang et al., who obtained efficient Brillouin amplification in gas-filled hollow-core waveguides [55], we increased the conversion efficiency using the capillary waveguides produced in translucent materials in the course of their multi-pulse drilling by the same discharge-pumped KrF laser. The nonlinear SBS or SRS gain is $G=PL_{eff}/\pi a^{2}$, where 2*a* is the diameter of a focal spot for a Gaussian beam, *L_eff_* is the effective nonlinear interaction length, which, for focusing in a free space, is doubled by the Rayleigh length $L_{eff}=2Z_{R}=2\pi a^{2}/\lambda$. This gives that G=2P/λ does not depend on focusing conditions for a given laser power *P* and wavelength *λ*. Compared with a free space, the radiation focusing into the capillary waveguide allows for the increase in *L_eff_* and keeping high laser intensity along the capillary length *L*, provided the radiation losses are small enough. The use of heavy fluorocarbon liquids inside the capillary is promising and has been demonstrated to act as excellent nonlinear media [52,56]. In addition, they have a refraction index approaching that of translucent dielectrics, which tend toward low losses in a capillary waveguide.

Another application of the temporally profiled laser pulse could be electron beam acceleration in a long, rippled capillary-like channel produced in PMMA by a high-power KrF laser where a long 100 ns pedestal forms the channel [33] while a high-powered final spike should produce the direct acceleration of the plasma’s electrons.

## 6. Conclusions

Our experiments on the multi-pulse drilling of polymethyl methacrylate and K8 silica glass by a discharge-pumped 20 ns rep-rate KrF laser revealed the self-focusing of UV radiation along the limited penetration depth is just a mechanism that manages filament-assisted ablation in deep capillary channels. In more detail, it proceeds through the following steps:A multiple specular reflection of the incident laser beam forms the initial cone-like ablative crater in the irradiated material that pre-focuses the radiation at the cone apex;The densification of the material by both UV radiation and ablation pressure causes a local increase in the refractive index ahead of the ablation front and initiates self-focusing of the UV radiation within a shallow penetration depth of UV light;Self-focusing breaks a laser beam into a multitude of self-organized filaments carrying power *P_f_* approximately equal to critical filamentation power being estimated as *P_cr_* ~ 2.5 × 10^4^ W, which is by an order of magnitude less than that of Kerr self-focusing;The peak fluence in filaments with the diameter 10–20 μm *F_f_* = 150–600 J/cm^2^ and corresponding peak intensity *I_f_* = (0.75–3.0) × 10^10^ W/cm^2^ are much higher than the threshold fluence *F_th_* for material ablation; such filaments of the bundle drill the sample in parallel; thus, the rate of filamentation-assisted ablation is independent of the initial pulse energy (or power);The exponential attenuation of radiation along the channel reduces the number of filaments in proportion to decreasing power while the maximal channel length logarithmically depends on the incident energy (power);In the course of radiation propagation in longer channels with the length *L* ≥ 10 mm, a single waveguide mode appears favorable for less radiation loss in the channel.

A similar drilling mechanism was shown to be realized in highly transparent media, e.g., KU-1 glass with a multiphoton absorption switched on instead of linear absorption. Waveguide laser beam propagation in long capillary channels was considered for direct electron acceleration by temporally profiled laser pulses with a high-powered final spike and efficient nonlinear compression of the excimer laser pulses into the picosecond range.

## Figures and Tables

**Figure 1 materials-15-08347-f001:**
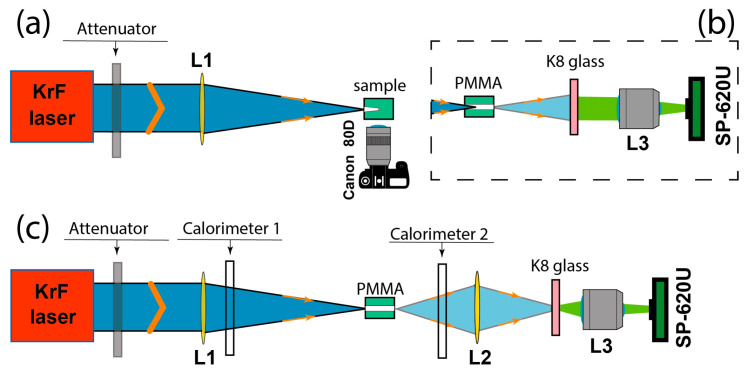
The layout of experiments on (**a**) multi-pulse sample drilling; (**b**) measurement of radiation distribution behind the drilled channel; (**c**) measurements of transmitted energy and distribution at the channel outlet.

**Figure 2 materials-15-08347-f002:**
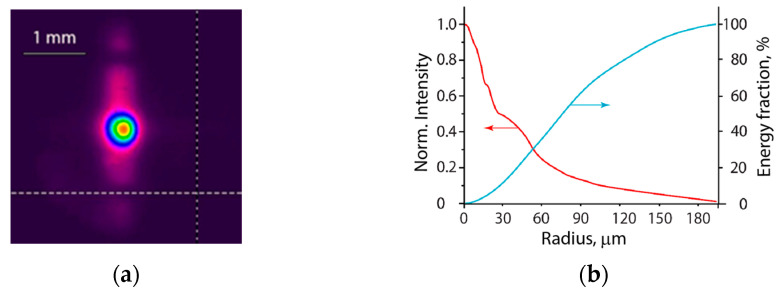
(**a**) Beam profiler image of the focal spot obtained with UV converter and (**b**) normalized distributions of intensity (red curve) and energy fraction (blue curve) in the focal spot.

**Figure 3 materials-15-08347-f003:**
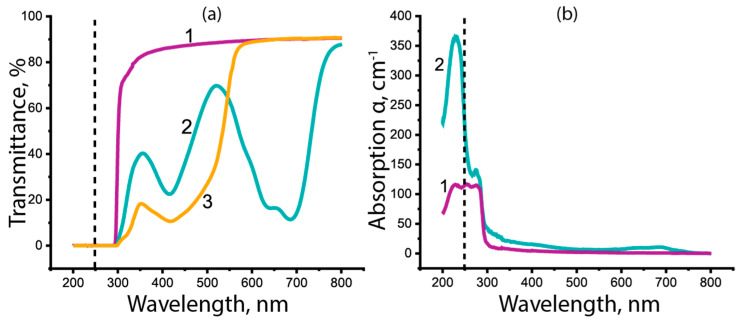
(**a**) Transmission spectra of (1) colorless PMMA of 1 mm thickness, (2) green PMMA of 3 mm and (3) yellow PMMA of 4 mm; (**b**) absorption coefficients in (1) colorless and (2) green PMMA. A dashed line designates the KrF laser wavelength.

**Figure 4 materials-15-08347-f004:**
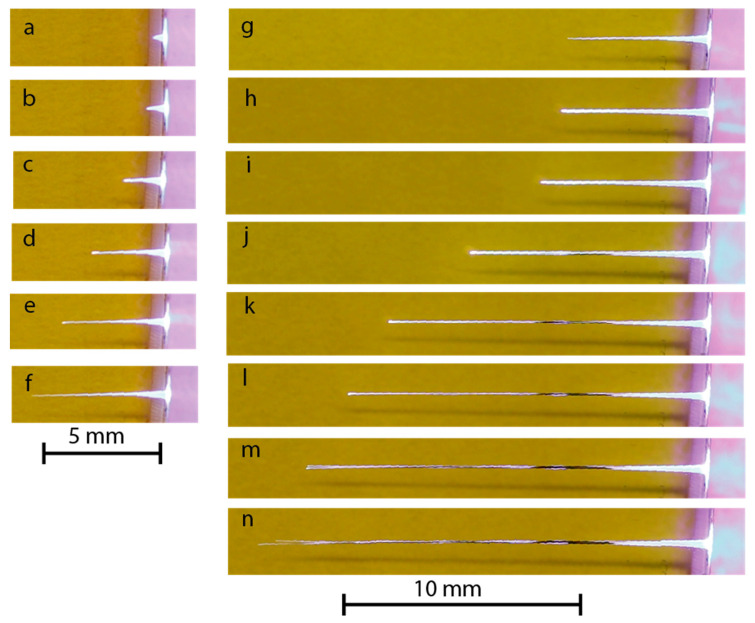
Video frame sequence of the growing channel in PMMA for the pulse energy (**a**–**g**) *E_L_* ≈ 2.8 and (**h**–**n**) 7 mJ. The laser beam falls on the sample from the right.

**Figure 5 materials-15-08347-f005:**
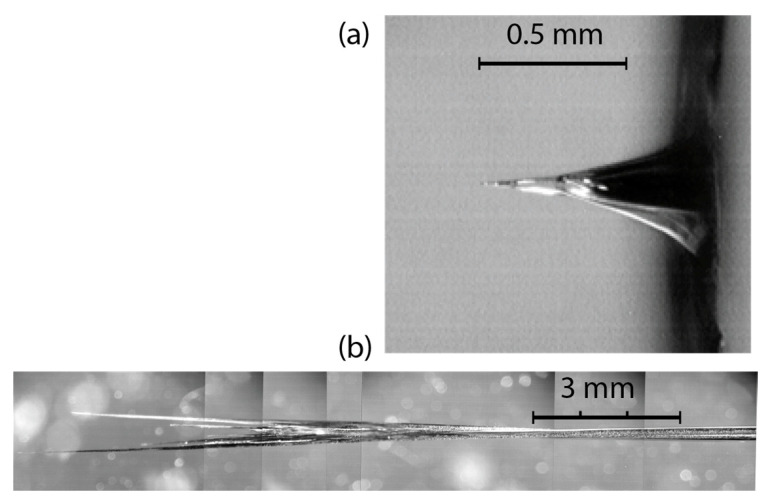
Microscopic images of final channels produced in PMMA by a 10 Hz train of pulses with (**a**) *E_L_* = 0.3 mJ and (**b**) 22.3 mJ (a fragment of the channel end is shown). The laser beam falls on the sample from the right.

**Figure 6 materials-15-08347-f006:**
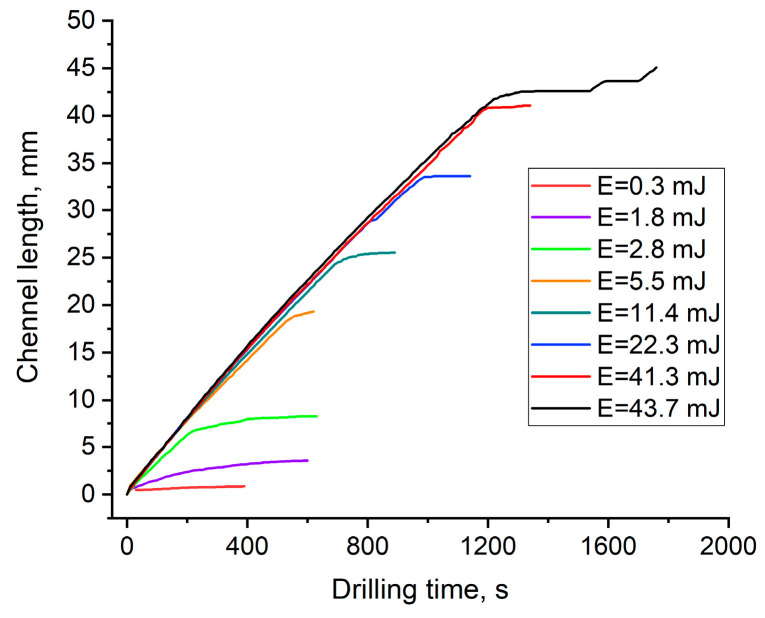
Channel length *z* vs. drilling time *t* for various pulse energies at a 10 Hz rep rate.

**Figure 7 materials-15-08347-f007:**
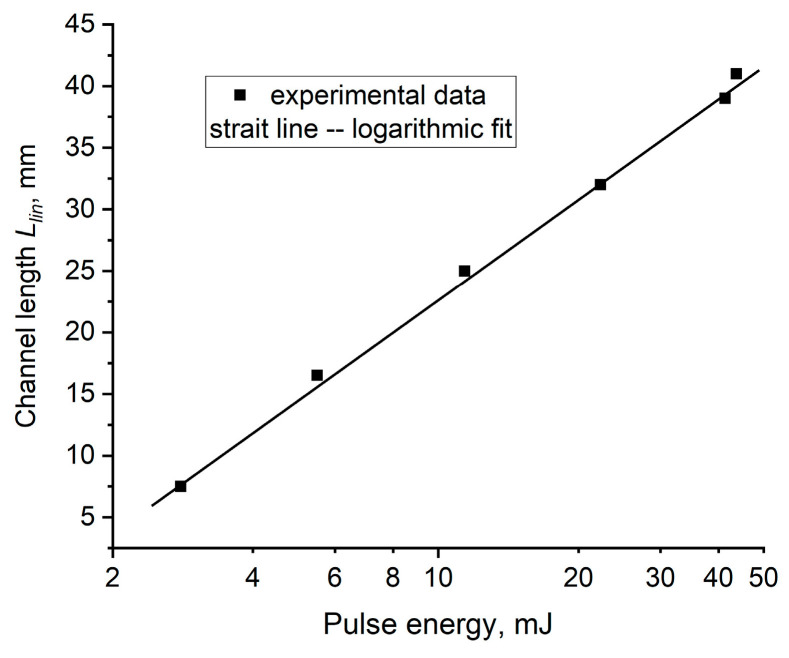
Channel length *L_lin_* in dependence on incident pulse energy.

**Figure 8 materials-15-08347-f008:**
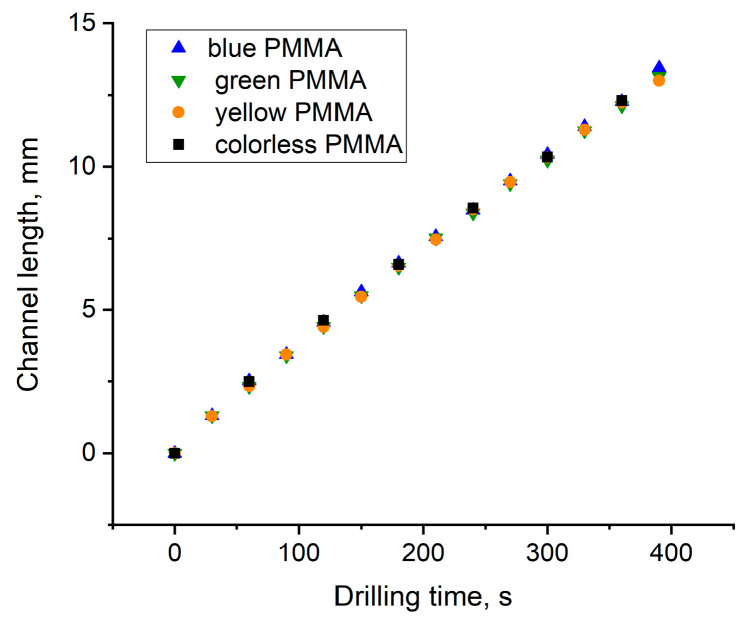
Channel length *z* vs. drilling time *t* for colorless and dyed PMMA samples at 10 Hz rep rate before saturation.

**Figure 9 materials-15-08347-f009:**
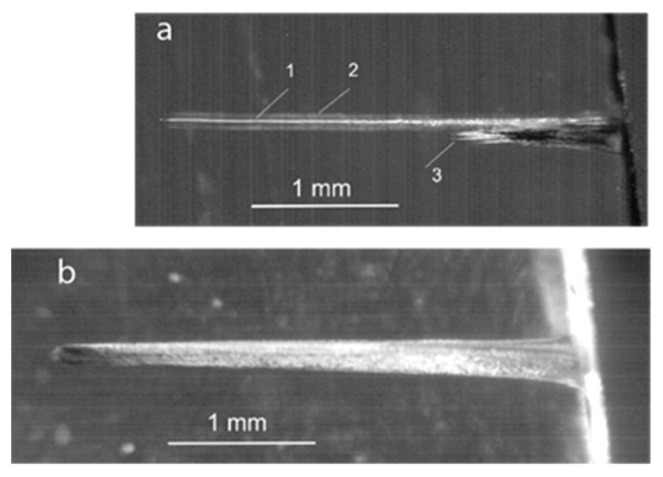
Microscopic images of channels drilled into K8 glass (**a**) by a train of pulses with *E_L_* = 2–4 mJ at a rep rate of 10 Hz for ~300 s, and (**b**) with *E_L_* ≈ 10 mJ at 40 Hz for ~ 180 s (not completed). The laser beam falls on the sample from the right.

**Figure 10 materials-15-08347-f010:**
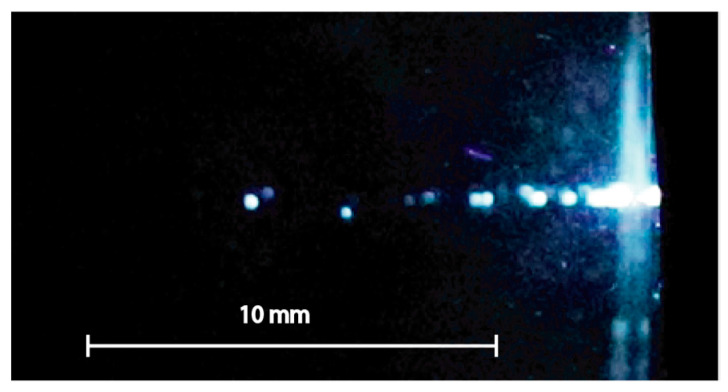
A video frame of the optical breakdown in transparent KU-1 glass at a pulse energy *E_L_* ≈ 2 mJ. The laser beam falls on the sample from the right.

**Figure 11 materials-15-08347-f011:**
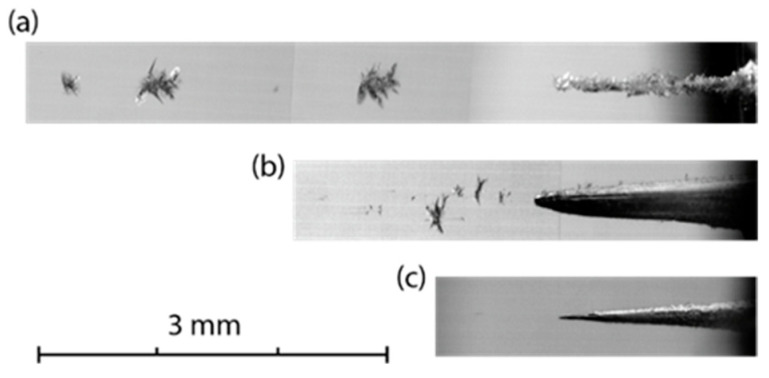
Microscopic images of the breakdown damage and completed channels produced in KU-1 glass by a rep-rate irradiation at 10 Hz with pulse energies (**a**) *E_L_* ≈ 2.1; (**b**) *E_L_* ≈ 5.1; (**c**) *E_L_* ≈ 22 mJ. The laser beam falls on the sample from the right.

**Figure 12 materials-15-08347-f012:**
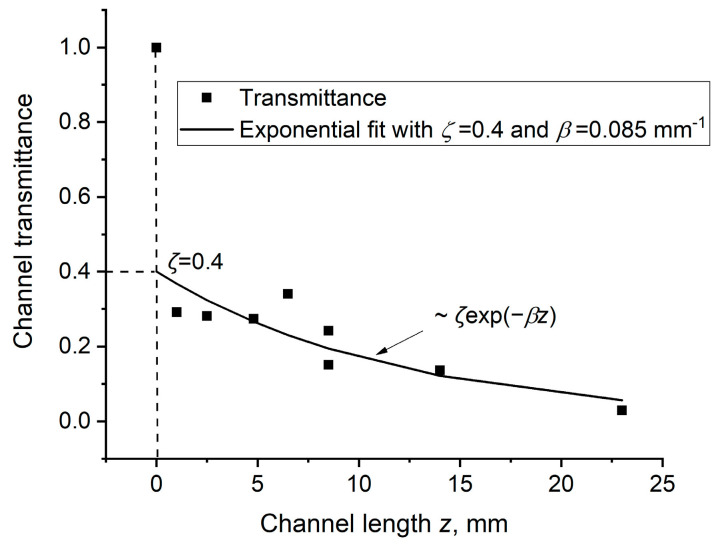
Channel transmittance and its dependence on PMMA thickness. The dots are measurements with calorimeters; the solid line is an exponential approximation with *ζ* = 0.4 and *β* = 0.085 mm^−1^.

**Figure 13 materials-15-08347-f013:**
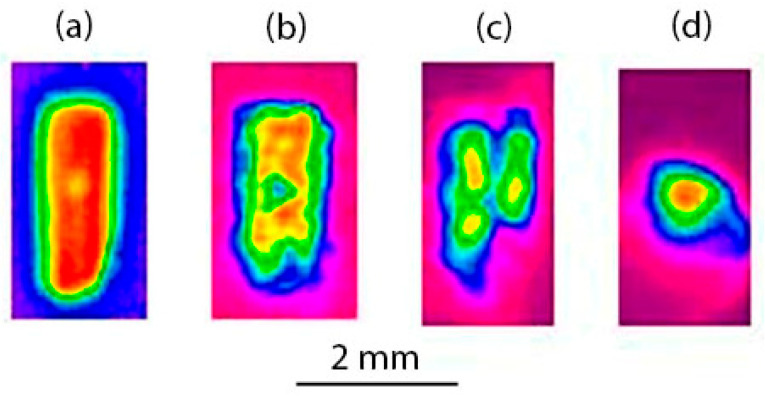
Profiler images of the fluorescence of UV-to-green converter placed at a distance of 100 mm (**a**) behind the lens focus and outlets of the through-channels of various lengths (**b**) 4.8, (**c**) 6.5 and (**d**) 23 mm. Pulse energy is ~15 mJ.

**Figure 14 materials-15-08347-f014:**
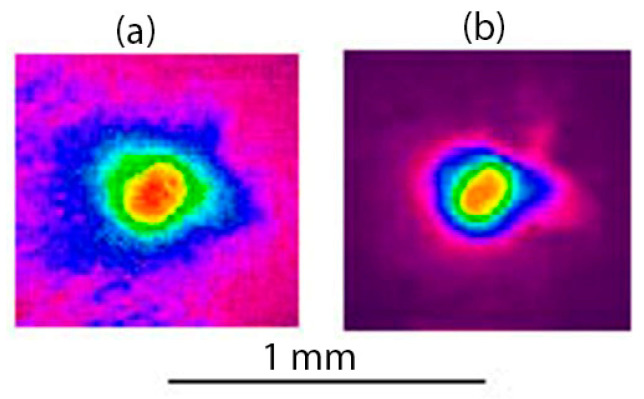
Distribution of laser radiation at the outlet of the channel of 23 mm length at different pulse energies (**a**) 110 and (**b**) 11 mJ.

## Data Availability

Not applicable.

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
