# Peer review of "A New Insight into High-Aspect-Ratio Channel Drilling in Translucent Dielectrics with a KrF Laser for Waveguide Applications"

_materials, 2022, doi:10.3390/ma15238347_

Round 1
Reviewer 1 Report
This paper discussed a new insight on capillary channel formation with a high aspect ratio in translucent matter by nanosecond UV laser pulses, provide additional arguments in favor of the decisive role of self-focusing in multi-pulse moderate-power ablation of translucent materials and demonstrate a waveguide nature of laser radiation transport within long channels.
There are some problems, which must be solved before it is considered for publication. If the following problems are well-addressed, this reviewer believes that the essential contribution of this paper are important for the interaction of UV laser radiation with polymers.
1. The INTRODUCTION does not clarify the research background of the work done in this paper, for example, the main work of this paper is multi-pulse drilling, but the first paragraph of the introduction does not mention drilling, but introduces surface ablation first; it does not cite enough previous work, and lacks an overview of the current development level and problems of the research; it spends a long space on the theoretical analysis of the research in this paper, which should be explained in a separate paragraph of theoretical analysis after the INTRODUCTION and before the EXPERIMENTAL SETUP;
2. The lack of logic for the introduction of the experimental setup. The overall system arrangement should be introduced first, and then a detailed description of the single components and light sources; in addition, such as in subsection 3.3.1, lines 320 and 321, "...... as shown in Figure 1", should be described clearly in advance in the description of Figure 1;
3. The two pictures in Figure 3 are not uniform in style and should both be labeled with the material indicated by each color line in the figure, or neither;
4. In subsection 3.2.1, the presentation of experimental results is loosely structured and lacks logic; consideration should be given to reordering, e.g., placing Figure 4 and Figure 8 together and presenting the experimental results in detail followed by an analysis of their causes;
5. The logical relationship between the experiment and the new insights presented in this paper is not clear enough in the discussion section. The significance of this paper is not expound sufficiently, the author need to highlight this paper's innovative contributions. A large amount of space is spent on introducing the self-focusing and Kerr effects in CONCLUSIONS, which should be clarified in the theoretical analysis chapter before the experiments.
Author Response
Please, see the attachment

Reviewer 2 Report
The authors proposed a new method on capillary channel formation with a high aspect ratio in translucent matter 9 by nanosecond UV laser pulses is discussed on the base of our experiments on KrF laser multi-pulse 10 drilling of polymethyl methacrylate and K8 silica glass. They indeed showed some laser drilling examples and provided detailed measurements on such as drilling depth v.s. laser energy. However, I don't see how they can ascribe these experimental results to the reason they claimed, material densification by both UV and ablation pressure followed by filamentation-assisted ablation. They just provided several possible mechanisms but didn't try to verify which was the dominant one in their experiments. I will agree to accept the publication of this paper if they can clarify this.
Author Response
Please, see the attachment

Round 2
Reviewer 1 Report
The authors have made moderate revisions to the article, which have resulted in a good improvement of both logic and readability. However, the presentation of the new insights presented in this paper could still be further improved, for example, in the DISCUSSION section, the conclusions of previous works, such as those in Refs. 25-28, 36-39, that the densification effect under UV and/or ablation pressure was clearly identified to increase a refraction index of the loaded material, could be briefly introduced to draw out the innovations of this paper. Furthermore, it is suggested that the most important conclusion presented in this paper, i.e., that it is just the self-focusing mechanism that manages filament-assisted ablation in deep capillary channels, be placed at the top of the CONCLUSION section.
Author Response
Please, see the attachment
